# The Potential Involvement of an ATP-Dependent Potassium Channel-Opening Mechanism in the Smooth Muscle Relaxant Properties of *Tamarix dioica Roxb.*

**DOI:** 10.3390/biom9110722

**Published:** 2019-11-10

**Authors:** Syeda Madiha Imtiaz, Ambreen Aleem, Fatima Saqib, Alexe Nicolae Ormenisan, Andrea Elena Neculau, Costin Vlad Anastasiu

**Affiliations:** 1Faculty of Pharmacy, Bahauddin Zakariya University, Multan 60800, Pakistan; Mahroshshah05@gmail.com (S.M.I.); ambreenaleem@hotmail.com (A.A.); fatima.saqib@bzu.edu.pk (F.S.); 2Department of Engineering and Management of Food and Tourism, Faculty of Tourism, Transilvania University, Brasov 500036, Romania; nickorme@unitbv.ro; 3Department of Fundamental, Prophylactical and Clinical Disciplines, Faculty of Medicine, Transilvania University, Brasov 500019, Romania; 4Department of Surgical and Medical Specialities, Faculty of Medicine, Transilvania University, Brasov 500019, Romania; canastasiu@gmail.com

**Keywords:** *Tamarix dioica*, HPLC-DAD, antidiarrheal, spasmolytic, bronchodilator, vasodilator, KATP channel opener

## Abstract

**Background**: *Tamarix dioica* is traditionally used to manage various disorders related to smooth muscle in the gastrointestinal, respiratory, and cardiovascular systems. This study was planned to establish a pharmacological basis for the uses of *Tamarix dioica* in certain medical conditions related to the digestive, respiratory, and cardiovascular systems, and to explore the underlying mechanisms. **Methods**: A phytochemical study was performed by preliminary methods, followed by HPLC-DAD and spectrometric methods. In vivo evaluation of a crude hydromethanolic extract of *T. dioica* (TdCr) was done with a castor-oil-provoked diarrheal model in rats to determine its antidiarrheal effect. Ex vivo experiments were done by using isolated tissues to determine the effects on smooth and cardiac muscles and explore the possible mechanisms. **Results**: TdCr tested positive for flavonoids, saponins, phenols, and tannins as methanolic solvable constituents in a preliminary study. The maximum quantity of gallic acid equivalent (GAE), phenolic, and quercetin equivalent (QE) flavonoid content found was 146 ± 0.001 μg GAE/mg extract and 36.17 ± 2.35 μg QE/mg extract. Quantification based on HPLC-DAD (reverse phase) exposed the presence of rutin at the highest concentration, followed by catechin, gallic acid, myricetin, kaempferol, and apigenin in TdCr. In vivo experiments showed the significant antidiarrheal effect of TdCr (100, 200, and 400 mg/kg) in the diarrheal (castor-oil-provoked) model. Ex vivo experiments revealed spasmolytic, bronchodilatory, and vasorelaxant activities as well as partial cardiac depressant activity, which may be potentiated by a potassium channel opener mechanism, similar to that of cromakalim. The potassium channel (K_ATP_ channel)-opening activity was further confirmed by repeating the experiments in glibenclamide-pretreated tissues. **Conclusions**: In vivo and ex vivo studies of *T. dioica* provided evidence of the antidiarrheal, spasmolytic, bronchodilator, vasorelaxant, and partial cardiodepressant properties facilitated through the opening of the K_ATP_ channel.

## 1. Introduction

The World Health Organization (WHO) claims that about one-third of the world’s population depends on a traditional system of medicines (mostly herbs) for their healthcare. Plants and herbs are the oldest companions of mankind. Plants are important not only for nutrition and shelter, but also in the treatment of various diseases. Herbal/traditional and/or natural medicines existed in various cultures and civilizations, like the Chinese, Egyptians, Unani/Tibb (South Asia), Western(allopathic), Kampo (Japan), and Greco-Arab. The system of modern medicine has now started to use botanicals after their scientific validation. A few famous examples like garlic, ginger, ginseng, gingko, and ispaghula are becoming popular among modern physicians. Likewise, studies on biological aspects of medicinal plants and the impact of the journals publishing this research are increasing at a rapid pace [1].

*Tamarix dioica* Roxb. ex Roth (Tamaricaceae) is locally known by various vernacular names like ghaz, jhao, khagal, lai, pilchi, and Ukkan in Pakistan [2,3]. *Tamarix dioica* is a perennial plant with the ability to tolerate changes in climate. It can grow in salty soil and dry places [4].

*T. dioica* is useful in the treatment of diarrhea, dysentery, and inflammation. It is also used for cold, fever, flu, and cough and as an astringent in burns, being leucodermic [5] and styptic in nature [6]. A paste of the dried bark and leaves has soothing effects on wounds [7,8]. *T. dioica* is also active against various serious bacterial and viral infections like tuberculosis, leprosy, gonorrhea, ringworm, polio, and measles [9,10].

Previous studies have suggested the presence of remarkable antifungal activity [11] and potent ulcer-protective and anti-inflammatory activities in *T. dioica* [12]. *T. dioica* leaves showed significant activities such as hepatoprotective, anti-oxidant/free radical scavenger [13,14], cytoprotective against gastric epithelial cell damage [5], and antimicrobial [15]. Furthermore, some species of the genus *Tamarix* have been previously reported to have cardioprotective potential, e.g., *Tamarix aphylla* [16].

Phytochemical studies confirmed the occurrence of flavonoids, phenols, proteins, amino acids, saponins, tannins, terpenoids, and fixed oils in aqueous extracts of various parts of the plant, as well as pholbatannins and steroids [2]. Spectroscopic analysis revealed the isolation of many flavonoids from *T. dioica* (aerial parts), like tamarexetin, tameridone, tamadone with hexacosyl pcoumarate, nevadensin A and apigenin, kaempferide, quercetin, and gardenins A, B, C, and E [17,18]. Additionally, D-mannitol (a sugar alcohol) and icteren (an inorganic compound) have been isolated [6,17].

Regardless of the availability of data about the traditional uses of *Tamarix dioica* in gastrointestinal, respiratory, and cardiovascular systems, the pharmacological basis for its use is scarce. The present study was performed to establish phytochemical studies as well as the pharmacological potential of *Tamarix dioica* Roxb. in treating the gastrointestinal, respiratory, and cardiovascular systems.

## 2. Materials and Methods 

### 2.1. Plant Collection, Extraction, and Fractionation

*Tamarix dioica* (aerial part) was collected in April 2016 from the university sites. The plant was identified by a taxonomist from the biology department of the same university. The voucher specimen “Stewart 489(6)” was submitted to the herbarium of the same department. The plant was cleaned carefully to get rid of any dirt and shade-dried until the grinding process took place. Later, the dried parts of plant were coarsely grinded by using an electronically operated mill. After grinding, approximately 1.0 kg of coarsely powdered *T. dioica* was soaked in hydromethanol (80% methanol + 20% distilled water) at room temperature for 10–12 days in a dark-colored bottle. The solution was filtered first through a cotton cloth and then through filter paper [19]. The solvent was dried using a rotary evaporator, Rotavapor (model 9230, BUCHI, labortechnik AG, Flawil, Switzerland) at reduced pressure until a dusky brown paste of a methanolic crude extract of *Tamarix dioica* (TdCr) formed with a yield of 13%, stored in an open-mouthed container at −35 °C.

For fractionation, 10 g TdCr was dissolved in 100 mL distilled water in a separating funnel and then shaken with dichloromethane (an organic solvent). The mixture was allowed to separate into two layers, which were then obtained and placed in different containers [19] The dichloromethane layer was dried on a rotary evaporator, Rotavapor (model 9230, BUCHI, labortechnik AG, Flawil, Switzerland) whereas the aqueous layer was dried through freeze-drying (Labcono Freezone 6L, Kansas, MO, USA) to obtain the dichloromethane (TdDcm) and aqueous (TdAq) fractions of TdCr with an approximate yield of 15% and 35%, respectively.

### 2.2. Animals and Their Housing

Experimental animals used were locally bred albino rabbits (1.0–2.0 kg, ♂/♀) and albino rats (150–250 g, ♂/♀). Animals were housed at room temperature under a controlled environment and given unrestricted access to food and tap water. Animals were starved overnight before the start of the experiment, with access to water only.

The experiments were completed according to the rules of the Institute of Lab Animal Resources, Commission on Life Sciences, National Research Council [20] and were permitted by the ethical committee of Bahauddin Zakariya University, Multan (EC/07 PHLP/2016) dated 3 March 2016.

### 2.3. Chemicals

All the chemicals were 99.9% pure and of research-grade quality. Acetylcholine, carbachol, doxazosin, glibenclamide, isoprenaline, loperamide hydrochloride, phenylephrine, and potassium chloride were obtained from Sigma Chemical Co. (St Louis, MO, USA), while cromakalim was procured from Tocris, Ellisville, MN, USA. Castor oil was purchased from Karachi Chemicals (Karachi, Pakistan). The chemicals used to make the physiological solutions, i.e., Krebs solution and Tyrode’s solution, were procured from Merck (Dermstadt, Germany) and BDH (Poole, England).

### 2.4. Phytochemical Screening

#### 2.4.1. Primary Phytochemical Examination

The hydromethanolic extract of *T. dioica* was examined qualitatively to determine the presence of functional compounds [21].

#### 2.4.2. Total Phenolic Content

Phenols were assessed by a method established earlier [22], using the Folin‒Ciocalteu (FC) reagent and gallic acid as the standards. TdCr (20 µL) was incubated for 30 min with FC (90 µL) before adding 6% sodium carbonate (90 μL). The mixture was kept at 37 °C for half an hour and the absorbance was noted at 630.0 nm in a microplate reader using the reagent as the blank. A similar procedure was carried out for the standard. All the experiments were performed in three replications. The total phenolic content (TPC) was stated as μg GAE/mg of crude extract.

#### 2.4.3. Total Flavonoid Content 

Total flavonoid content (TFC) determination of *T. dioica* was done by a colorimetric method using aluminum chloride (AlCl_3_) as the base [22]. A mixture comprising 20 μL of TdCr, 10 μL of potassium acetate, 10 μL of aluminum chloride, and 160 μL of distil. water was incubated at room temperature for half an hour. The absorbance of the reaction mixture was read at 415.0 nm by a microplate reader (Sterile transparent 96 well plate, SPL life science, Gyeonggi-do, Korea). TFC was stated as μg QE/per mg of extract after triplicate analysis.

#### 2.4.4. HPLC-DAD Analysis

Stock solutions of reference standards, i.e., apigenin, caffeic acid, catechins, gallic acid, kaempferol, myricetin, quercetin, and rutin, were made and then diluted with methanol to reach a final concentration of 50 μg/mL. Methanol was used as the solvent for polyphenols at a concentration of 10 mg/mL. The solutions were sonicated and filtered using a sartolon polyamide membrane filter Sartorious (Sonicator, Sweepzone technology, Kearny, New Jersey, USA) (0.2 μM). Samples were made fresh before analysis or kept at 4 °C if not to be tested for more than 1 h.

The HPLC-DAD study of the hydromethanolic extract of *T. dioica* for polyphenols was done by following an established method [23]. HPLC (HPLC-DAD, 1200 series, Agilent technologies, Waldbronn, Germany) was prepared with a C18 analytical column (250.0 mm, 4.60 mm and 5.0 μM) coupled with a diode array DAD detector. For polyphenols, two mobile phases were used; mobile phase ‘A’ is comprised of acetonitrile, methanol, water, and acetic acid (5:10:85:1), while mobile phase ‘B’ contains acetonitrile, methanol, and acetic acid (40:60:1). The rate of flow was adjusted to 1.2 mL/min. A 20-μL sample of the solution dissolved in methanol was inserted into the column, and the column was reconditioned every 10 min early on in the next analysis. The absorption was read at 257.0 nm, 279.0 nm, 325.0 nm, and 368.0 nm for rutin, gallic acid, catechin, caffeic acid, myricetin, quercetin, and kaempferol.

### 2.5. In Vivo Experiments

#### Antidiarrheal Activity in Rats

Antidiarrheal activity was studied through some modifications of a previous method [24]. Thirty rats were divided into control, standard, and test groups (*n* = 6) and fasted for 24 h prior to testing with access to water. Animals were caged singly creased with filter paper. Animals in Group I (the control) were administered normal saline (1 mL/kg), Group II (the standard) were administered loperamide (3 mg/kg), and Groups III–V were treated with various doses (100, 200, or 400 mg/kg) of crude extract of *T. dioica*. Doses were selected based on initial trials, and three increasing doses were given. After 1 h, 1 mL of castor oil was given orally to all groups. Animals were observed for about 4 h for watery diarrhea.
% inhibition = (A ‒ B/A) × 100
A = wet stool count in control group 
B = wet stool count in treated group

### 2.6. Ex Vivo Experiments Using Isolated Tissues

Rabbits are commonly used as laboratory animals for isolated tissue experiments. Several tissues can be isolated from the same animals to make their use more economical and reduce ethical objections. Ex vitro experiments were performed according to the previously described methods, with minor modifications [25]. All the contractile activities were recorded using force transducers (MLT 0015, isotonic and FORT-100, isometric transducers) attached to a PowerLab data acquisition system (AD Instruments, Sydney, Australia), which was attached to a PC with Labchart software (AD Instruments, Sydney, Australia) version 8.

#### 2.6.1. Rabbit Jejunum Preparation

Isolated jejunum preparation was used to evaluate the mechanism of antidiarrheal activity. an isolated jejunum of about 2.0 cm in length was held in a tissue organ bath of 15 mL containing Tyrode’s salt solution of a standard composition. The tissue baths were gurgled with carbogen (95% O_2_ & 5% CO_2_ and kept at a temperature of 37 °C. The tissues were permitted to adjust for 30 min before the start of the experiment and then stabilized with acetylcholine (ACh, 0.3 μM); the normal Tyrode solution was substituted before beginning the experiment until a similar pattern was observed [25].

To explore the mechanism of the relaxation, the crude extract was examined on continual spasmodic contractions provoked by K^+^ (25 mmol/L) as well as K^+^ (80 mmol/L), which are considered useful for the identification of any inhibitory mechanisms like calcium channel blocking and/or potassium channel activator mechanism. TdCr was administered to the induced sustained contractions to examine the inhibitory responses. The observed relaxant activity of TdCr was represented as a percentage of the control contraction. An agent that completely reduces the contractions provoked by K^+^ (25 mmol/L) only is denoted as a potassium channel activator, whereas agents that relax both K^+^ (25 mmol/L) and K^+^ (80 mmol/L) are referred to as calcium channel blockers. This test discriminates conclusively between calcium channel blockers and potassium channel activators [26]. Pretreatment of tissue with glibenclamide (K_ATP_ channel antagonist) reduces the concentration‒response curve of TdCr against K^+^ (25 mmol/L), which shows the contribution of the K^+^ ATP channel opener activity.

#### 2.6.2. Rabbit Tracheal Preparation

The trachea of a rabbit was separated out and placed in Krebs solution of a standard composition. Isolated tissue was sited in a 15-mL organ bath bubbled constantly with carbogen, kept at the temperature of 37 °C, and equilibrated for 1 h prior to the beginning of the experiment. The tissue was further stabilized with carbamylcholine (1 µmol/L) to get a similar response. TdCr was applied after the contractions provoked by carbamylcholine (1 µmol/L) and K^+^ (25 mmol/L) to find the anticholinergic and potassium channel-opening potential. The standards were also tested on carbamylcholine, and K^+^ (25 mmol/L) provoked contractions for the comparative study and determination of mechanism of activity [25].

#### 2.6.3. Rabbit Aortic Ring Preparation

The vasoconstrictor/vasorelaxant effect of TdCr was checked using isolated aortic preparation: the descending aorta of a rabbit was obtained after dissection and kept in a Krebs physiological salt solution. A 2–3-mm-wide sample of aortic tissue was cut and placed in a 15-mL organ bath bubbled with carbogen at 37 °C. After 1 h of equilibration, TdCr was added to check for the presence of a vasodilator/vasoconstrictor effect [25].

#### 2.6.4. Isolated Paired Atria

Spontaneously contracting heart muscle was removed from a rabbit subsequent to euthanasia by cervical dislocation by a method previously described [27]. The atria were carefully removed, hung in a tissue organ bath with the left atrium underside, and placed in Krebs solution with a supply of carbogen. The rate and force of atrial contractions were recorded. The tissue was left for 30 min to stabilize, with regular changes of fluid at 10-min intervals. The cardiodepressant/-stimulant effect of TdCr was evaluated by cumulative addition of it to the tissue in an isolated organ bath.

#### 2.6.5. Statistical Analysis

The values of spontaneous and induced contractions in isolated tissue preparations (ex vivo studies) were expressed as mean ± standard error mean (mean ± SEM), with the median effective concentrations (EC50) values at 95% confidence intervals. The data on in vivo antidiarrheal activity were analyzed by one-way ANOVA and Dunnett’s t-tests. Values of 0.05 and *p* < 0.005 were considered significantly and most significantly different values, respectively. All the calculations, graphing, and statistical analysis were performed using GraphPad Prism version 6 (San Diego, CA, USA).

## 3. Results

### 3.1. Phytochemical Analysis

#### 3.1.1. Preliminary Phytochemical Study

The preliminary qualitative analysis confirmed the existence of flavonoids, phenols, saponins, and tannins in TdCr.

#### 3.1.2. Total Flavonoid Content

The test revealed that the total flavonoid content in TdCr was 36.17 ± 2.36 μg QE/mg extract.

#### 3.1.3. Total Phenolic Content

The estimation of the total phenolic contents in TdCr highlighted the presence of 145 ± 0.002 μg GAE/mg extract.

#### 3.1.4. HPLC-DAD Analysis

Chromatography by HPLC-DAD is a well-known, simple, sensitive, reproducible, and dependable technique for chemical screening. The HPLC profile of the methanolic extract of *T. diocia* was compared with the retention time (RT) and UV spectrum of reference compounds including polyphenols, i.e., gallic acid, rutin, kaempferol, apigenin, quercetin, caffeic acid, catechin, and myricetin (Table 1). The reverse-phase (RP) HPLC fingerprinting of a methanolic extract of *Tamarix dioica* has confirmed the presence of gallic acid (0.882 µg/mg), catechin (2.02 µg/mg), rutin (2.29 µg/mg), apigenin (0.417 µg/mg), myricetin (0.82 µg/mg), and kaempferol (0.5 µg/mg) in a crude extract of *Tamarix dioica*; the highest quantity found was of rutin, at 2.29 µg/mg, while caffeic acid and quercitin were under the lower limit of detection (LOD). The maximum amount (2.02 µg/mg) of catechin was detected in the sample that was compared with the standard and it was found that the retention time remained almost the same. A minimal amount of apigenin (concentration: 0.41 µg/mg was found (Figure 1).

### 3.2. In Vivo Experiments

#### Antidiarrheal Activity in Rats

The methanolic extract of *T. dioica* had a significant antidiarrheal effect when orally administered at doses of 100.0, 200.0, and 400.0 mg/kg. It exhibited prominent inhibition of loose stools compared to the control group of 13.62% (*p* < 0.05), 45.0% (*p* < 0.01) and 60.62% (*p* < 0.005), respectively. The positive control, loperamide (3 mg/kg), caused a 74.1% (*p* < 0.005) inhibition, which was found to be the maximum. The control saline group (NS, 10 mL/kg) caused negligible inhibition of diarrhea. The results were statistically evaluated by ANOVA (one-way) and multiple comparison by Dunnett’s test and presented in a bar graph (Figure 2).

### 3.3. Ex Vivo Experiments

#### 3.3.1. Rabbit Jejunum Preparation

The hydromethanolic extract of *T. dioica* led to complete relaxation when administered to the impulsive (natural) contractions of an isolated preparation of jejunal tissue at 0.3–3.0 mg/mL (EC50 = 0.334 mg/mL; CI 95%: 0.224–0.516) (Figure 3a and Figure 4a). TdCr, when applied to contractions provoked by K^+^ (25 mmol/L), led to relaxation having EC50 = 0.5 mg/mL; CI 95%: 0.39–0.65, while partial relaxation was observed against K^+^ (80 mmol/L)-provoked contractions at 10 mg/mL (Figure 4a, Figure 5a and Figure 6a). The response was similar to that of cromakalim, a K_ATP_ channel opener, which relaxed K^+^ (25 mmol/L)-provoked contractions (EC50 = 0.029 mg/mL; CI 95%: 0.010–0.181). When its response on K^+^ (25 mmol/L)-provoked contractions was studied again after the application of glibenclamide (GB, 1 µmol/L), a K_ATP_-channel blocker, significant inhibition was seen and the result was (EC50 = 1.06 mg/mL; CI 95%: 0.405–2.755) vs. (EC50 = 0.60 mg/mL; CI 95%: 0.35–0.42). The observed relaxing effect of TdCr was almost blocked similar to cromakalim in glibenclamide-pretreated tissue (Figure 6a,b).

Furthermore, the organic (dichloromethane) fraction of TdCr (TdDcm) showed relaxant/spasmolytic action on both, the spontaneous contractions (EC50 = 0.31 mg/mL; CI 95%: 0.174–0.524) (Figure 4b) and K^+^ (25 mmol/L) as well as on K^+^ (25 mmol/L)-provoked contractions (EC50 = 0.28 mg/mL; CI 95%: 0.166–0.486) (Figure 4b), whereas TdAq showed complete relaxation of natural contractions at 3 mg/mL (EC50 = 0.361 mg/mL; CI 95%: 0.192–0.660) and K^+^ (25 mmol/L)-provoked contractions were relaxed at 10 mg/mL (EC50 = 2.83 mg/mL; CI 95%: 1.833–4.355) (Figure 4c).

#### 3.3.2. Isolated Tracheal Strip Preparation

TdCr and its fractions were investigated for a possible bronchodilatory effect by applying it to contractions provoked by K^+^ (25 mmol/L), K^+^ (80 mmol/L), and carbamylcholine (1 µmol/L) in an isolated tracheal preparation. TdCr caused the relaxation of K^+^ (25 mmol/L) and CCh (1 µmol/L)-provoked contractions (with respective values of EC50 = 0.49 mg/mL; CI 95%: 0.381–0.657, and 1.80 mg/mL; CI 95%: 1.407–2.306), with only partial relaxation of contractions provoked by K^+^ (80 mmol/L) at the maximum concentration of 10 mg/mL. The results indicated that TdCr has greater potency against contractions provoked by K^+^ (25 mmol/L); furthermore, it was found that the relaxation was entirely suppressed in the presence of 3 µmol/L glibenclamide (EC50 = 0.13 mg/mL; C.I 95%: 0.031–0.401), and the same was true of cromakalim (EC50 = 0.013 mg/mL; C.I 95%: 0.01–0.135) (Figure 7a,b).

A dichloromethane fraction of TdCr (TdDcm) led to the relaxation of both K^+^ (25 mmol/L) and carbamylcholine (1 µmol/L) (EC50 = 0.36 mg/mL; C.I 95%: 0.085–1.503 and EC50 = 0.41; C.I 95%: 0.26–0.62)(Figure 8a), while an aqueous fraction, i.e., TdAq, applied to tracheal tissue led to the relaxation of carbamylcholine-provoked contractions (EC50 = 0.23 mg/mL; C.I 95%: 0.16–0.32) Moreover, K+ (25 mmol/L)-provoked contractions were relaxed at 10 mg/mL (EC50 = 17.85 mg/mL; C.I 95%: 6.61–48.28) (Figure 8b).

#### 3.3.3. Isolated Preparations of Aortic Ring

TdCr showed vasodilation after being applied to K^+^ (25 mmol/L) and PE (1 µmol/L)-provoked contractions (calculated EC50 = 0.70 mg/mL; CI 95%: 0.505–0.952, and 4.00 mg/mL; CI 95%: 2.164–7.413), while partial relaxation was found after exposure to K^+^ (80 mmol/L) (Figure 9a). The relaxation of K^+^ (25 mmol/L) was nearly completely blocked by glibenclamide (3 µmol/L) pretreatment (EC50 = 0.12 mg/mL; CI 95%: 0.025–0.355), and the same was true of cromakalim (EC50 = 0.01 mg/mL; CI 95%: 0.02–0.277) (Figure 10).

The dichloromethane fraction of TdCr (TdDcm), on exposure to K^+^ (25 mmol/L), showed a significant relaxant response (EC50 = 0.023 mg/mL; CI 95%: 0.02–0.03) (Figure 9b), while PE (1 µmol/L) provoked contractions (EC50 = 0.857 mg/mL; CI 95%: 0.58–1.22) (Figure 9b). The polar fraction of TdCr (TdAq) revealed relaxation of K^+^ (25 mmol/L) contractions at a concentration of 0.1–10 mg/mL and PE (1 µmol/L)-provoked contractions relaxed fully at a concentration of 0.03–1.0 mg/mL. Both fractions were found to be devoid of any noteworthy effects on K^+^ (80 mmol/L)-provoked contractions (Figure 9c).

#### 3.3.4. Effects on Paired Atrial Preparation

On application, TdCr produced a partial cardiodepressant effect in a paired atrial preparation from a rabbit. A negative inotropic effect was produced (EC50 = 0.31 mg/mL; CI 95%: 0.031–0.732), as well as a negative chronotropic effect (EC50 = 2.07 mg/mL; CI 95%: 0.220-4.041). The effect was compared to cromakalim, which produced negative inotropic and chronotropic effects (EC50 = 0.40 mg/mL; CI 95%: 0.029–0.893 and EC50 = 2.0 mg/mL; CI 95%: 0.269–4.410, respective) (Figure 11 and Figure 12).

## 4. Discussion

The experiments were planned to investigate the pharmacological action of *Tamarix dioica* to scientifically prove its traditional uses. Phytochemical studies were done by a preliminary estimation of the total content of phenols and flavonoids as well as HPLC-DAD. The phytochemical studies of a methanolic crude extract of *T. dioica* showed the presence of flavonoids, phenols, saponins, tannins, and terpenoids. HPLC-DAD examination of the methanolic plant extract was performed to explore the presence of therapeutically useful plant constituents. Flavonoid glycosides (rutin and kaempferol), flavone aglycones (apigenin and catechins), flavanone (myricetin), and polyhydroxy phenolic compound (gallic acid) were quantified in a crude extract of *Tamarix dioica*. Phenols and flavonoids have been previously reported to be related to various types of pharmacological and oxidative action in biological systems [28].

The confirmation of these plant metabolites in the extract establishes a parallel correlation regarding the significant bioactivity of the plant extract, e.g., rutin has immense pharmacological potential in terms of its antiplatelet aggregating, antioxidant, antispasmodic, cytoprotective, antihypertensive, and anti-inflammatory effects, as well as the prevention of gastric ulceration based on animal models [29]. Gallic acid is reported as a scavenger of free radicals and inducer of apoptosis in many cancer types like lung cancer, colon adenocarcinoma cell lines, etc. It performs an important role in delaying malignancy as well as the development of cancer. Catechins are proanthocyanins belonging to the flavones class and are known to possess antioxidant properties. Kaempferol belongs to the flavanols and has anti-ulcer, anti-inflammatory, and anti-oxidant properties. Myricetin is a flavanone possessing antioxidant, anticarcinogen, and antimutagenic properties. Apigenin is a 4/,5,7trihydroxyflavone belonging to the flavones class; it is a potent cell proliferation inhibitor, good for bone health, and possesses antioxidant, anti-inflammatory, antidepressant, and cardioprotective effects. It can be deduced from the present results that the notable antispasmodic, antihypertensive, and cardioprotective potential of *T. dioica* might be ascribed to rutin, quercetin, and apigenin, respectively [30,31,32].

The hydromethanolic extract of *T. dioica* (TdCr) was tested to evaluate its possible therapeutic potential for managing various human diseases. To evaluate the traditional use in the treatment of diarrhea, *T. dioica* was assessed for its antidiarrheal potential using a castor oil (olio di ricino)-provoked diarrheal model in rats. TdCr significantly controlled (*p* < 0.05) the diarrhea at the given doses of 100, 200, and 400 mg/kg in rats. It is well known that castor oil provokes diarrhea because it has ricinoleic acid as its active component, which inhibits the transportation of water and electrolytes [33]; as a result, the motility of the colon is increased, which results in diarrhea [34]. Hence, it can be supposed that the antidiarrheal effect of TdCr was due to an antimotility and/or antisecretory mechanism, using loperamide as a standard. The antidiarrheal potential of most medicinal plants was previously found to be due to the existence of alkaloids, flavonoids, saponins, sterols, tannins, and/or terpenes as constituents. Previously reported data suggested that flavonoids may inhibit the intestinal secretory response provoked by the prostaglandin E2 [35]. Furthermore, flavonoids exhibited antioxidant properties, which are suspected to be responsible for the inhibitory responses exerted upon numerous enzymes, including those involved in the metabolism of arachidonic acid. As the phytochemical screening showed that flavonoids, tannins, and terpenoids are found in a methanolic extract of *T. dioica*, this might be a reason for its antidiarrheal activity [36].

To explore the mechanism of antidiarrheal activity, TdCr was further investigated through ex vivo experiments using isolated rabbit jejunum tissues. TdCr induced the spasmodic effect in a concentration-dependent style on an isolated preparation of jejunum; this inhibitory effect suggested the existence of component(s) with a relaxing effect in the crude extract. Various bioactive components are present in plants that exert therapeutic and protective activities in numerous metabolic illnesses, particularly those related to the smooth muscles of the human body. Contractions in smooth muscles depend mainly on increasing cytoplasmic Ca^2+^ concentrations, either by Ca^2+^ ion channels or by receptor instigation (G-protein and/or voltage-gated). The intracellular Ca^2+^ ions allow it to bind with calmodulin, a protein. The resultant complex triggers both actin and myosin; the contractile proteins, because of phosphorylation of the myosin light chain by MLC-kinase and binding with actin, contribute to contraction [37]. The relaxation of smooth muscle is due to the elimination of the contractile stimulus or the inhibition of a mechanism responsible for contraction.

Furthermore, MLC phosphatase is responsible for the dephosphorylation of light chains, negative regulation of muscular contractions, and the reduction of Ca^2+^ sensitivity as well. So, all these mechanisms help in the relaxation of muscles (Figure 13) [38].

Previous studies revealed that the spasmolytic effect of medicinal plants may involve K^+^ channel opener activity, calcium channel blocker potential, and receptor mediation [39]. The mechanism of relaxation was confirmed by applying it to contractions provoked by K^+^ (25 mmol/L) and K^+^ (80 mmol/L). TdCr led to significant relaxation of contractions provoked by K^+^ (25 mmol/L) in a concentration-dependent manner, with only partial relaxation of K^+^ (80 mmol/L)-provoked contractions at the highest concentration. Substances that selectively inhibit the contractions provoked by K^+^ (25 mmol/L) are denoted as potassium channel activators/openers [40]. Whereas, on the other hand, substances that inhibit the contractions provoked by both doses of K^+^ (25 and 80 mmol/L) are termed Ca^2+^ channel blockers. These experiments effectually differentiate between potassium channel openers and calcium channel blockers from a mechanistic viewpoint.

The inhibitory effects of TdCr on contractions provoked by K^+^ (25 mmol/L) indicate its K- channel opener activity. The different types of K^+^ channels found in the human body include K_ATP_ channels, K^+^ Ca channels, delayed rectifier K^+^ channels, and inward rectifier K^+^ channels.

To explore the contributions of different types of potassium channels, TdCr was applied to K^+^ (25 mmol/L)-provoked contractions in glibenclamide-pretreated tissue, a known blocker of the ATP-dependent K^+^ channels [41]. The effect of TdCr on K^+^ (25 mmol/L) was diminished when applied in glibenclamide-pretreated tissue and was like that of cromakalim, a well-known K_ATP_ channel opener. Hence, it confirms the contribution of ATP-dependent potassium channel opener activity as a prime spasmolytic mechanism of TdCr.

In view of the traditional use of TdCr in respiratory disorders and the medicinal use of potassium channel openers in numerous respiratory disorders including cough and asthma, a crude methanolic extract of *T. dioica* was tested on isolated trachea for the possible presence of bronchodilatory effects. TdCr instigated a reduction of K^+^ (25 mmol/L) and CCh (1 µmol/L)-provoked contractions, being more potent against K^+^ (25 mmol/L) compared with CCh-provoked contractions. The results suggested the potassium channel opener activity of TdCr mediated by ATP-dependent potassium channels. Furthermore, the bronchodilatory potential in *T. dioica* may be due to the presence of potassium channel opener activity as well as flavanols and flavonoids among the plant constituents [42].

*T. dioica* was further investigated for any possible effects on the cardiovascular system. TdCr was applied on K^+^ (25 mmol/L), K^+^ (80 mmol/L), and PE (1 µmol/L)-provoked contractions in the isolated preparation of aortic rings to explore its possible vasodilatory effects. TdCr resulted in partial relaxation of K^+^ (80 mmol/L)-provoked contractions, with full relaxation of K^+^ (25 mmol/L) and PE (1 µmol/L)-provoked contractions. The relaxation of K^+^ (25 mmol/L)-provoked contractions showed significant inhibition in glibenclamide 3 µmol/L pretreated tissues, which confirms the contribution of K_ATP_ channels as a mechanism of vasodilation that can be used to treat hypertension [43].

The fractionation of TdCr results in a dispersal of components among the polar and nonpolar phases. Both the organic and aqueous fractions led to smooth muscle relaxation by potassium channel-opening activity. The organic fraction was more potent than the aqueous fraction.

In experiments to evaluate the medicinal potential in disorders related to the cardiovascular system, TdCr showed a cardiodepressant effect, affecting both the force and rate of contraction. The cardiodepressant effect was found to be similar to that of cromakalim, an K_ATP_ channel opener. The cardiodepressant effect of cromakalim might be due to decreasing the time of myocardial action potential and triggering hyperpolarization [44,45]. It has already been documented that cardiac cells are less sensitive to the K_ATP_ channel compared to smooth muscle cells [46]. Earlier studies revealed that the depressant effects of cromakalim on cardiac muscles take place at a 30‒100-fold higher concentration as compared to the relaxation of smooth muscles [47,48]. The involvement of K_ATP_ channels in the physiology of the heart is not yet established [49]; furthermore, different isoforms of K_ATP_ channels exists in cardiac and smooth muscles, which might help elucidate the comparatively lower inhibitory effect of TdCr on cardiac muscles [50].

## 5. Conclusions

This study determines the phytochemical profiling and antidiarrheal, spasmolytic, bronchodilatory, vasodilatory, and cardiodepressant activities of a crude methanolic extract of *Tamarix dioica*. The phytochemical profiling shows the presence of saponins, tannins, flavonoids, and phenols. The results of HPC-DAD revealed the presence of rutin in the highest concentration, followed by catechin, gallic acid, myricetin, kaempferol, and apigenin. In vivo and ex vivo studies demonstrate the presence of significant antidiarrheal, spasmolytic, bronchodilatory, vasodilatory, and partial cardiodepressant activities, possibly through the ATP-dependent potassium channel-opening property, providing the pharmacological basis for its traditional uses.

## Figures and Tables

**Figure 1 biomolecules-09-00722-f001:**
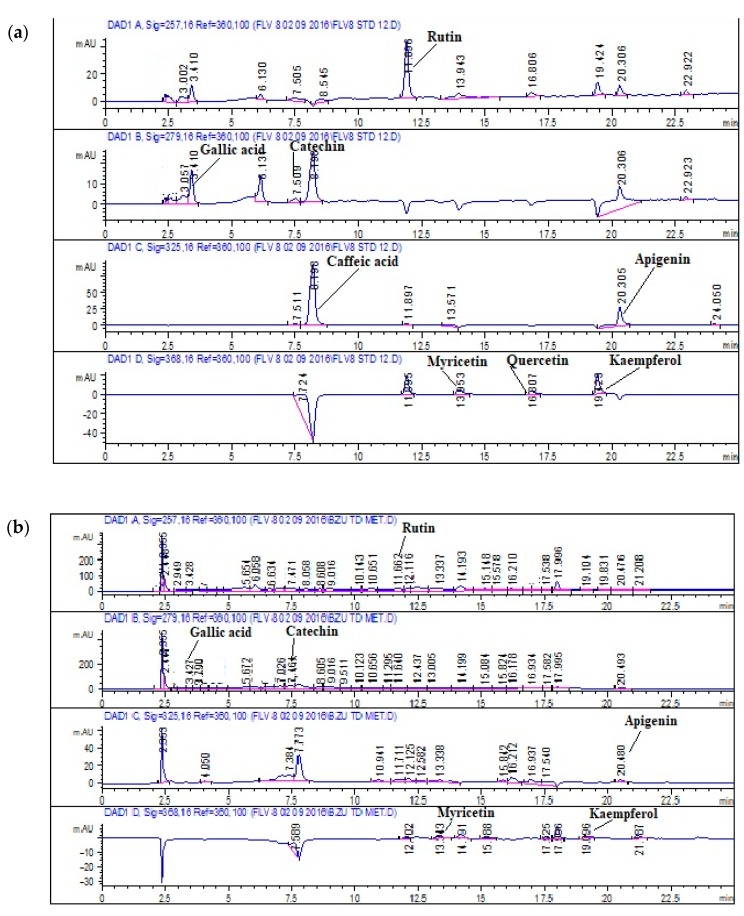
HPLC-DAD chromatograms of (**a**) Standard at 257, 279, 325, and 368 nm; (**b**) Methanolic extract of TdCr extract at 257, 279, 325, and 368 nm.

**Figure 2 biomolecules-09-00722-f002:**
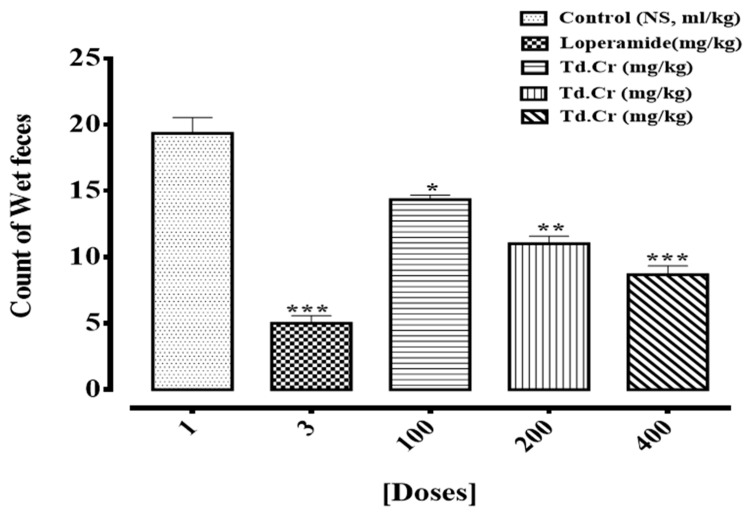
Bar graph representing the antidiarrheal activity of TdCr by a castor-oil-provoked method in rats. Bars express mean ± SEM with *n* = 6, analyzed by using ANOVA (one-way) and multiple comparison by Dunnett’s test (**p* < 0.05, ***p* < 0.01, ****p* < 0.005 when compared to the control).

**Figure 3 biomolecules-09-00722-f003:**
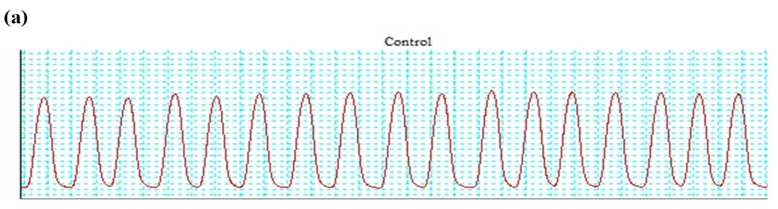
Tracing showing (**a**) Spontaneous contraction (Control) (**b**) The effect of methanolic extract of TdCr in comparison to the control on spontaneous contractions of isolated rabbit jejunum preparations.

**Figure 4 biomolecules-09-00722-f004:**
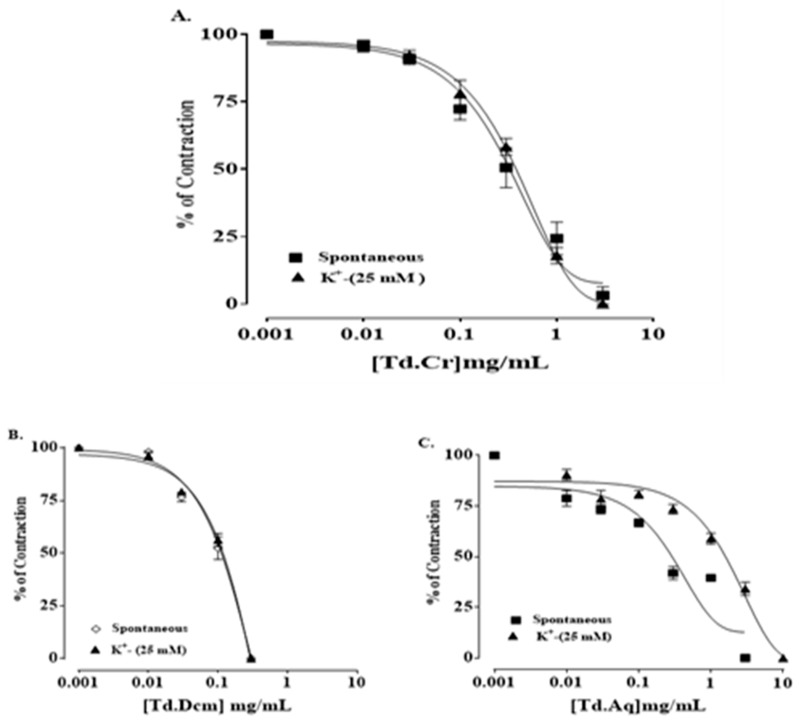
Concentration-response graphical presentation of (**A**) TdCr, (**B**) TdDcm, and (**C**) TdAq on natural and K+ (25 mmol/L) provoked contractions of isolated jejunal preparation. Values shown as mean ± SEM, *n* = 5.

**Figure 5 biomolecules-09-00722-f005:**
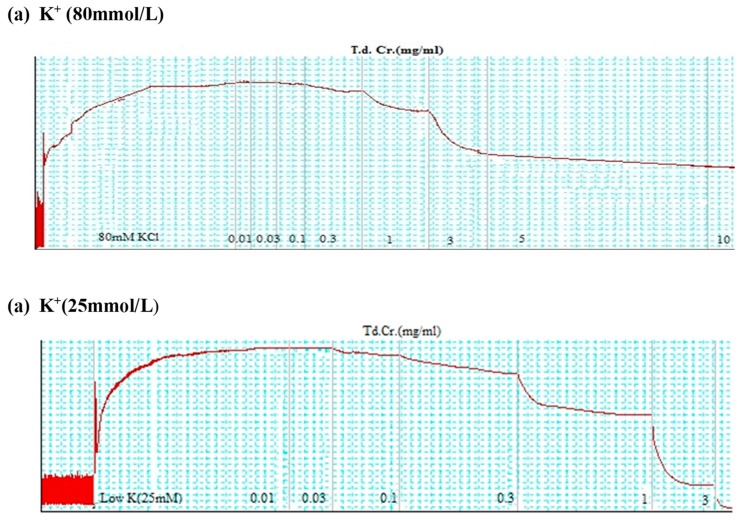
Tracings showing the effect of a methanolic extract of TdCr plant on (**a**) high K^+^ (80 mmol/L) and (**b**) low K^+^ (25 mmol/L)-induced contractions in isolated rabbit jejunum preparations.

**Figure 6 biomolecules-09-00722-f006:**
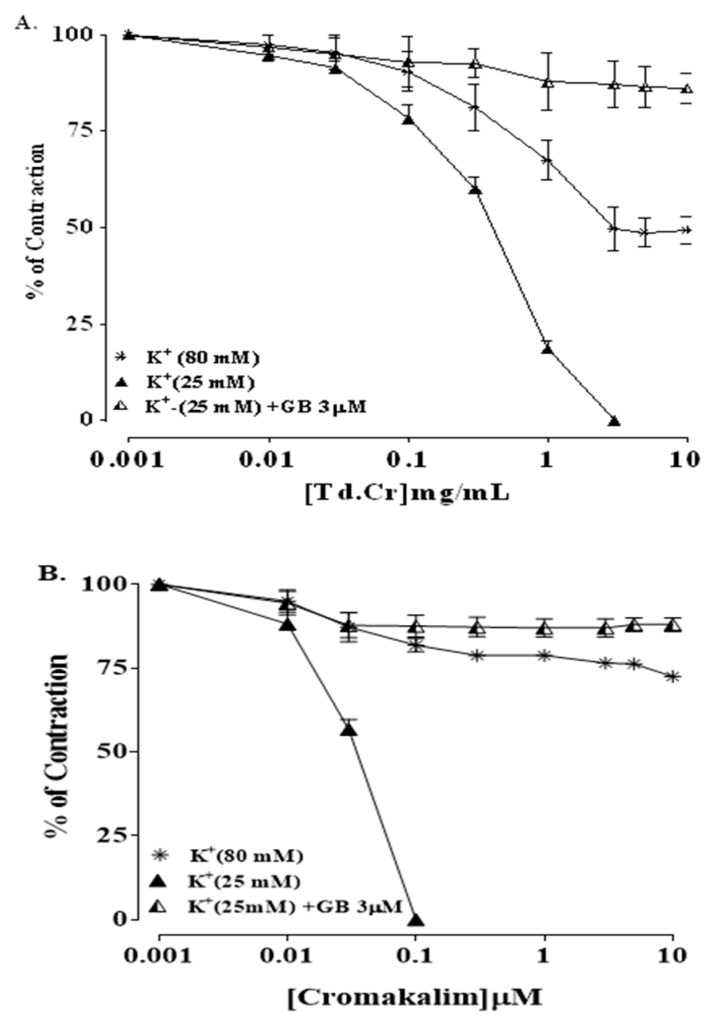
Concentration‒response graphical presentation of (**A**) TdCr compared with (**B**) cromakalim against K+ (80 mmol/L) in the absence or presence of glibenclamide (GB; 3 μmol) in isolated jejunal preparation. Values shown as mean ± SEM, *n* = 5.

**Figure 7 biomolecules-09-00722-f007:**
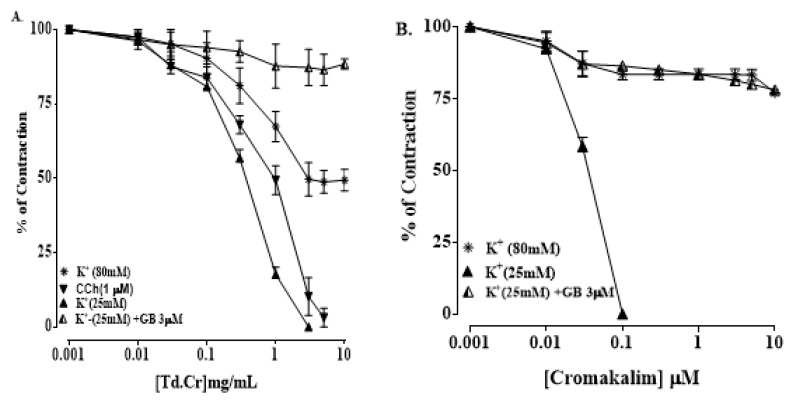
Concentration‒response graphical presentation of (**A**) TdCr compared with (**B**) cromakalim (1 µmol/L) on K+ (80 mmol/L) and K+ (25 mmol/L)-provoked contractions in the absence or presence of glibenclamide (3 μM) isolated rabbit tracheal preparations. Values shown as mean ± SEM, *n* = 5.

**Figure 8 biomolecules-09-00722-f008:**
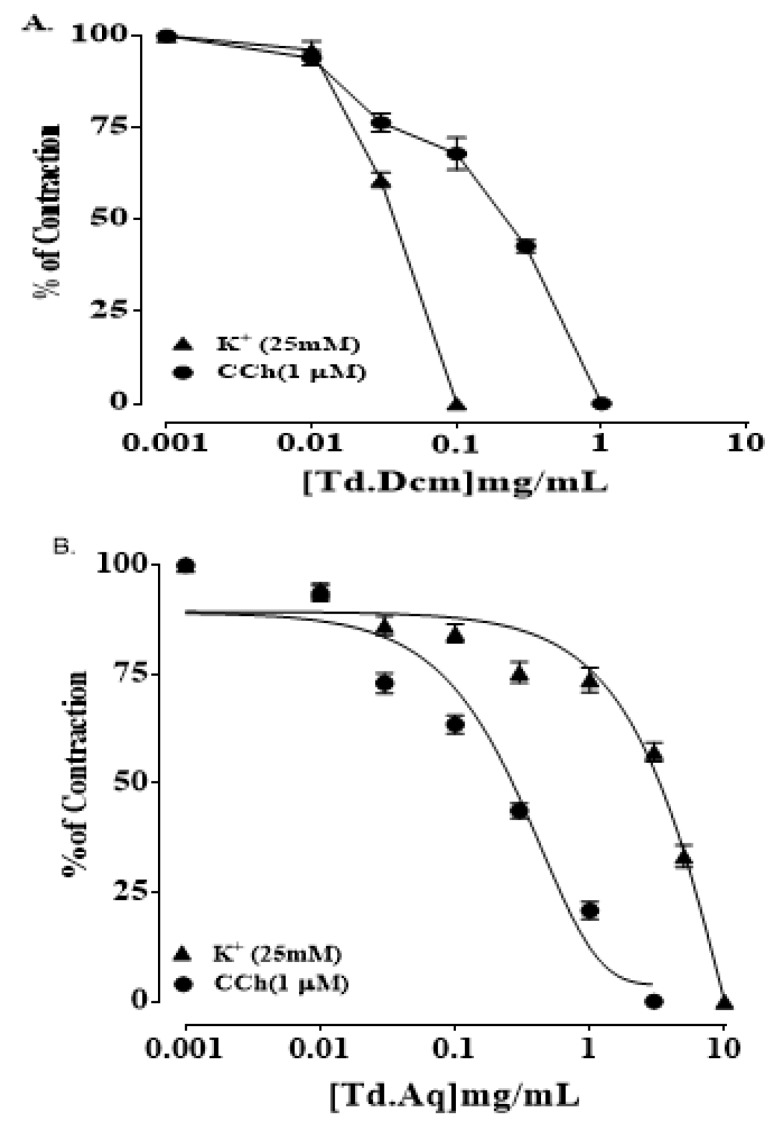
Concentration‒response graphical presentation of (**A**) TdDcm and (**B**) TdAq on contractions provoked by CCh (1 µmol/L) and K^+^ (25 mmol/L) in isolated rabbit tracheal preparations. Values shown as mean ± SEM, *n* = 5.

**Figure 9 biomolecules-09-00722-f009:**
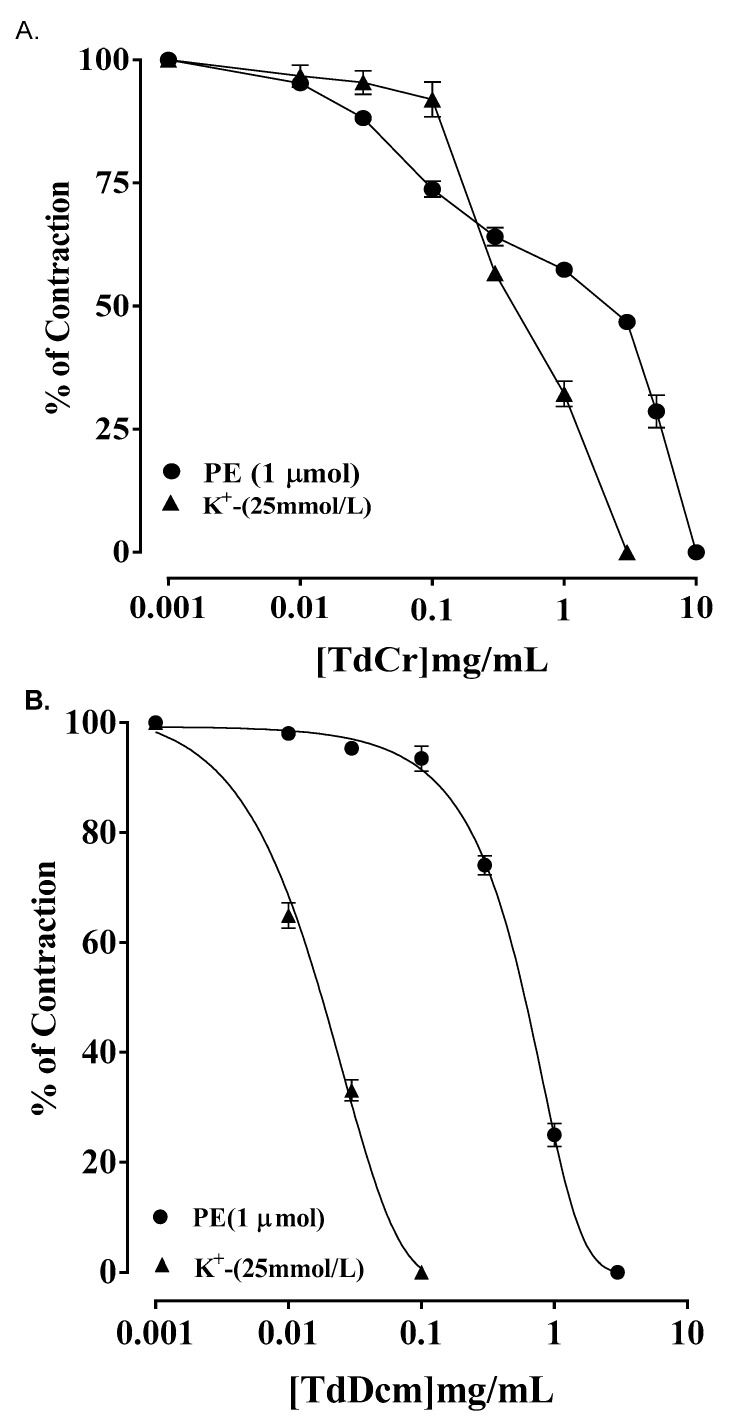
Concentration‒response graphical presentation of (**A**) (TdCr), (**B**) TdDcm, and (**C**) TdAq on K+ (25 mmol/L) and phenylephrine (1 µmol/L) provoked contractions of isolated aortic ring preparations. Values shown as mean ± SEM, *n* = 5.

**Figure 10 biomolecules-09-00722-f010:**
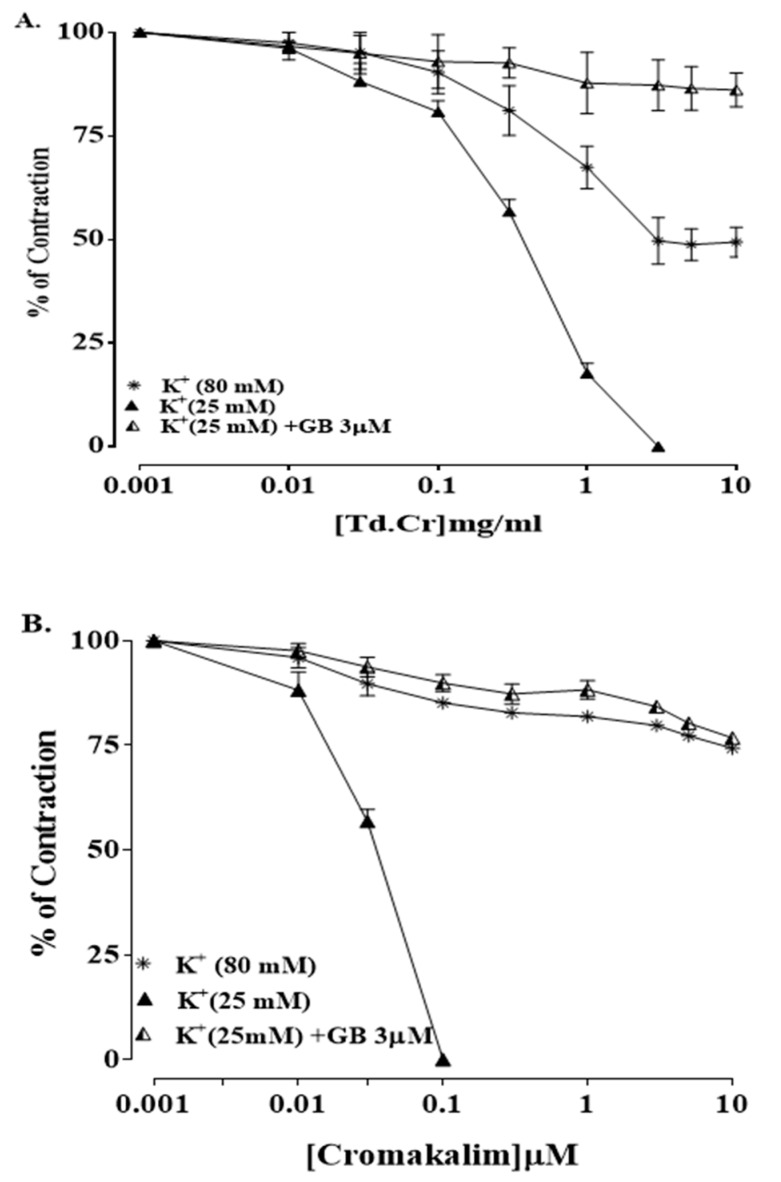
Concentration‒response graphical presentation of (**A**) TdCr in comparison to (**B**) cromakalim against contractions provoked by K^+^ (80 mmol/L) and K^+^ (25 mmol/L) in the absence or presence of glibenclamide (GB; 3 μM) in isolated aortic ring preparations. Values shown as mean ± SEM, *n* = 5.

**Figure 11 biomolecules-09-00722-f011:**
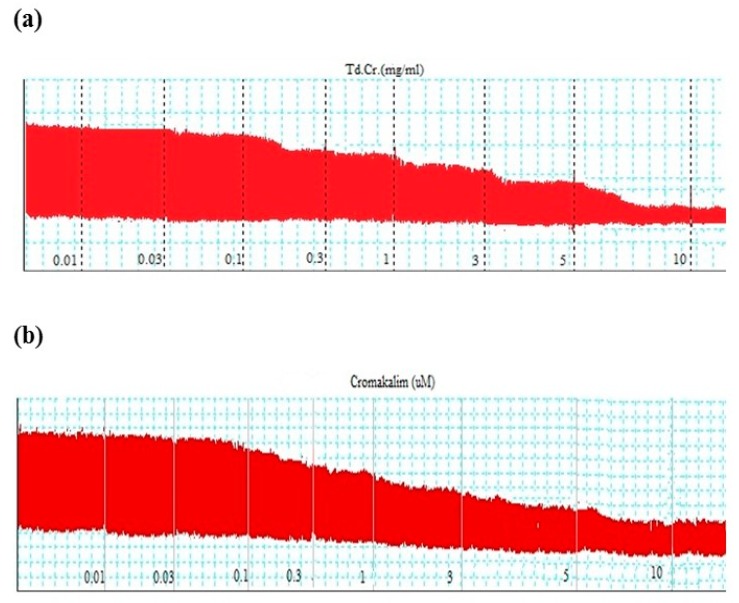
Tracings indicating the effect of (**a**) TdCr and (**b**) cromakalim on spontaneous contractions of paired atrial preparation of a rabbit.

**Figure 12 biomolecules-09-00722-f012:**
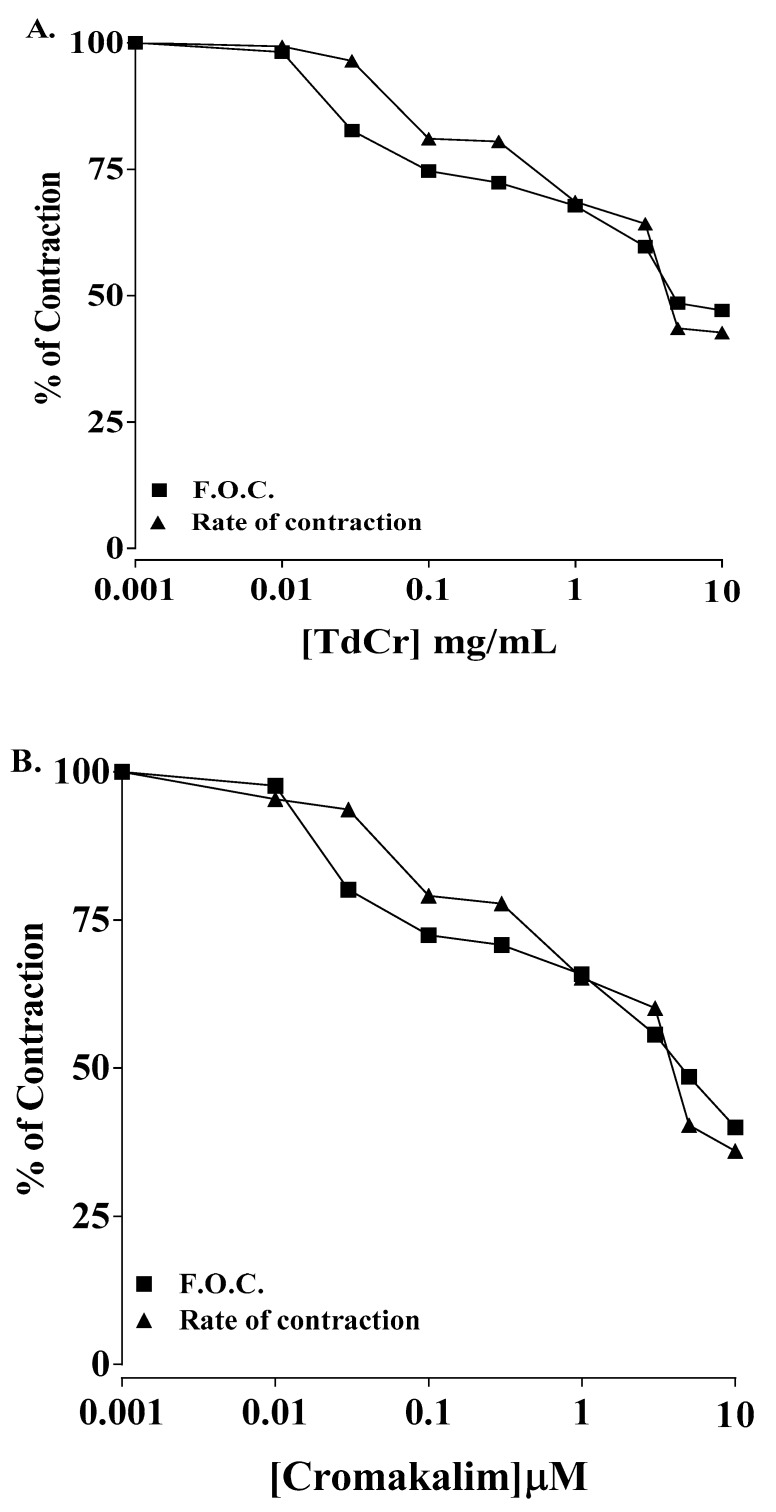
Concentration-response curve presentation the inotropic and chronotropic effects of (**A**) TdCr and (**B**) cromakalim on spontaneous contractions of paired atrial preparation of a rabbit. Values shown as mean ± SEM, *n* = 5.

**Figure 13 biomolecules-09-00722-f013:**
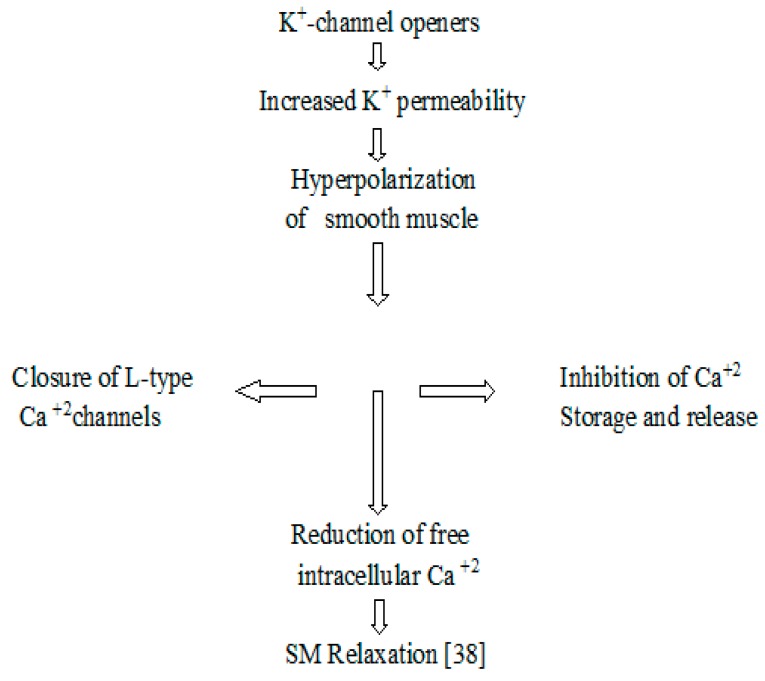
The schematic diagram of proposed mechanism of smooth muscle relaxation by K^+^ channel openers.

**Table 1 biomolecules-09-00722-t001:** Retention time of detected components in TdCr extract in comparison with reference flavonoids determined by HPLC-DAD analysis.

Compound	Signal Wavelength	Retention Time as Per TdCr (min)
Rutin	257	4.35
Gallic acid	257	15.91
Catechin	279	9.7
Caffeic acid	325	49.15
Apigenin	325	23.53
Myricetin	368	18.72
Quercetin	368	23.59
Kaempferol	368	21.31

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
