# Peer review of "The Potential Involvement of an ATP-Dependent Potassium Channel-Opening Mechanism in the Smooth Muscle Relaxant Properties of Tamarix dioica Roxb."

_biomolecules, 2019, doi:10.3390/biom9110722_

Round 1

Reviewer 1 Report

Authors present the pharmacological effect of T. dioica extracts. The extract content was defined and exhaustive in vivo and ex vivo examination was performed indicating the particular mechanisms of action in animals of obtained extracts. In my opinion the manuscript should be published in present form.

Author Response

Dear Sir                               

I am highly thankful to the reviewers for their precious time and invaluable comments for improving manuscript ID: biomolecules-631569 entitled, “Potential involvement of ATP-dependent potassium channel opening mechanism in smooth muscle relaxant prospective of Tamarix dioica Roxb. ”. We have carefully addressed all the comments which really helped in improving the manuscript. We have removed typo and grammar errors.The corresponding changes and refinements made with red color in the revised paper are summarized in our response below.

REVIEWER 1:

Authors are highly obliged for your precious time and effort to review the manuscript.

Reviewer 2 Report

In this manuscript the authors performed first the phytochemical characterization of TdCr (Tamarix dioca) used in Pakistan traditional medicine regarding to anti-inflammatory and anti-diarrheic properties. They identified by HPLC-DAD analysis the major flavonoid glycosides (rutin and kaempferol), flavone aglycones (apigenin and catechins) and polyhydroxy phenolic compound (gallic acid) constituents in the methanolic crude extract of T. dioica. By using an in vivo model of castor oil induced diarrhea on rats, they showed that TdCr significantly controlled the provoked diarrhea, a result similar to anti-diarrheic traditional use of the plant. By several ex vivo experiments on smooth muscle tissues (rabbit isolated jejunum, isolated aortic ring, tracheal strip preparation) and cardiac tissue (rabbit isolated paired atria), they demonstrated that the extract produced relaxing effects. They showed that these effects could be due mainly by increased potassium permeability. As the specific KATP channels blocker glibenclamide inhibited the relaxing effects of TdCr in their experimental conditions (pre-contracted preparations in 25 mM potassium external concentration), they identified this ionic channel to be a potential target of the crude methanolic extract of T dioica. They conclude that antidiarrheal, smooth muscle relaxation (spasmolytic, bronchodilator, vasodilator) and partial cardiodepressant activities of T. dioica through ATP dependent potassium channel opening property are providing the pharmacological basis of the traditional use of the plant.

This work is well performed and written; the manuscript seems to be well balanced and fair. It is carried out in a careful manner; to my mind this manuscript is acceptable for publication.

However there are some points that the authors should improved before.

Title:

Potential involvement of?

Abstract:

Lines 32-34: not clear

Line 37: “… of T dioica indicated the presence”: not appropriate

Introduction:

Lines 71-72: The authors are indicating that the study is performed to evaluate the pharmacological potential of T. dioica in the cardiovascular system: however, there is any description of the use of the plant in traditional medicine in the treatment of hypertension or cardiovascular disease in the paragraph connected to traditional use of the plant. Is there some data about potential cardiovascular effects described in the literature?... or is it the availability of vascular and cardiac tissues due to the use of rabbits to obtain jejunum that offers the opportunity to investigate the properties of T dioica on the cardiovascular axis?

Material and methods:

The authors used two animal models, rats for in vivo experiments and rabbits for ex vivo pharmacological studies. Why not only rats?

Did the authors try to investigate the ex vivo contractile properties of isolated duodenum or jejunum of the rats by using the model of diarrhea induction? It could be very interesting to compare the spontaneous contractile activity between the different groups, and also to verify if T dioica applied in the organ bath modifies the contractile activity of intestinal tissue of diarrhea induced rats versus control.

Line 157: ex vivo and not ex vitro

Line 158-159: please indicate the reference of ADI force tranducers

Ex vivo experiments: please indicate the temperature of organ bath

What is the number of rabbits used? Does the n indicated in results correspond to the number of animals or the number of preparations isolated from one animal?

Rabbit jejunum preparation:

The reference 24 does not seem to be appropriate there; this reference, a review, does not correspond to the experimental design described (lines 168-179). (see discussion)

Line 181: Krebs solution and not Kreb solution

Rabbit aortic ring preparation:

Line 192: a “2-3 mm aortic tissue was cut”: is that indicates that the authors only used a piece of 2-3 mm of one aorta of a rabbit? Or did they cut an aorta in several rings?

Results:

Figure 1: legend needs to be improved: it seems difficult to a non-specialist to read the chromatograms and to identify the position the major compounds. Please show on the chromatograms the position of each compound of interest.

Anti-diarrheal activity: figure 2: The control group concerns rats with castor oil induced-diarrhea, with approximately 20 wet stool. What is the count for non-induced rats?

Rabbit jejunum preparation: An example of a characteristic tracing of the concentration-dependent relaxing response of TdCr could be demonstrative, in the two conditions, spontaneously and pre-contracted model. Typical recordings should improve the manuscript.

The crude extract was examined on continual contractions induced by hyperK extracellular media but also on spontaneously contractile activity. Is there an effect not on only the amplitude of contraction-relaxation cycles but also on the frequency of these cycles?

Figure 3: n = 5? Please see methods. n is the number of rabbits used (N= 5 and the number of sample per animal is 1) or n=5 number of the samples per animal?

Effects on paired atrial preparations: An example of a characteristic tracing of the concentration-dependent cardiodepressant response of TdCr could be demonstrative. Typical recordings should improve the manuscript. What is the initial value of the spontaneous frequency of contraction of the isolated preparation?

Figure 9: FOC?

Discussion:

Could the authors compare the value of the EC50 of TdCr obtained in each model? Is it similar? Is there different efficiency between the tissues? As the main target of the compounds of T dioica seems to be KATP channels, is the expression of KATP channels in these tissues identical?

Line 405: the description of mechanisms linked to smooth muscle relaxation seems summarized there, in comparison to the description of the mechanisms involved in contraction, even if the authors present a graphical figure. MLC phosphatase and its regulation?

On the graphical figure, an arrow on the right indicates “inhibition of Ca2+ storage and release”: inhibition of Ca2+ storage induces an increase of free intracellular calcium!

The authors have characterized the main constituents of T dioica, for example rutin, with the highest amount. Did the authors investigate the effects of this compound in their experimental models in order to confirm that rutin could be relaxant on smooth muscle tissues and cardiodepressant in cardiac tissue? Could rutin or another major compound of T dioica be a KATP opener?

What are the immediate perspectives of this study? Do the authors expect to perform electrophysiological studies?

Author Response

Dear Sir                               

I am highly thankful to the reviewers for their precious time and invaluable comments for improving manuscript ID: biomolecules-631569 entitled, “Potential involvement of ATP-dependent potassium channel opening mechanism in smooth muscle relaxant prospective of Tamarix dioica Roxb. ”. We have carefully addressed all the comments which really helped in improving the manuscript. We have removed typo and grammar errors.The corresponding changes and refinements made with red color in the revised paper are summarized in our response below.

REVIEWER 2:

Title: The recommended change has been made in the title.

Abstract: The recommended changes in the abstract has been made.

Introduction:

The authors are indicating that the study…...… on the cardiovascular axis?

Reply: We selected this plant for the above said activities based on literature survey and known traditional uses of the plant Tamarix dioica. Data indicate that several species of Tamarix (e.g. Tamarix aphylla) have potential in cardiovascular disorders (Ashour et al., 2012). The data has been added in the manuscript too.

Material and Method:

The authors used two animal models……. Why not only rats?

Reply: Previous studies suggested that rabbit jejunum is considered as an appropriate tissue preparation to assess the contractile and/or relaxant effects because of spontaneous rhythmic contractions (Mehmood et al., 2011). While, Rodents both rats and mice are the most common used experimental animals (Ghayur and Gilani, 2005; Meiti et al., 2009; Nwidu et al., 2011).

Did the Author try to investigate …. Diarrhea induced rats versus control.

Reply: Initially, we performed in vivo experiments to check the anti-diarrheal potential of Tamarix dioica, then used isolated tissues to confirm and explore the possible mechanism.

Ex vivo not the ex vitro

Reply: The above said change has been made in the manuscript.

Please indicate the reference of ADI force transducer

Reply: Reference of force transducer has been added in manuscript.

Please indicate the temperature of organ bath

Reply: The temperature of organ bath was kept at 37 oC. Temperature is also added in manuscript as per suggested.

What is the number of rabbits used?.................... one animal.

Reply: For every isolated tissue experiment, we used different rabbits. So, n indicate number of rabbits used for that specific experiment.

The reference 24 does not seem…. Experimental design described

Reply: The above-mentioned reference has been replaced with the more appropriate one. Reference number 24 become 25 now because of addition of a reference.

Krebs solution not Kreb solution

Reply: The correction has been made in the manuscript

Rabbit aortic ring preparation

Reply:  A single ring of 2-3 mm of aortic ring has been made from one rabbit.

Results:

Figure1: …….

Reply: Legend has been improved. Chromatograms have been improved and the position of major compounds has been added as suggested by respected reviewer.

Anti-diarrheal activity: figure 2…..

Reply: No wet stool has been seen in non-induced rats. As wet stool is the indicator of diarrhea induction after administration of castor oil.

Rabbit jejunum preparation: An example of a characteristic tracing……

Reply: Tracings has been added as suggested

The crude extract was examined…….. frequency of these cycles?

Reply: Isolated jejunum and paired atria are spontaneously contracting preparations and effect can be observed on both the amplitude and the frequency of contractions. While the induced contractions of High K+, Low K+, and carbamylcholine are the sustained contractions and only changes in amplitude can be observed.

Figure 3: n=5…………….

Reply: In the above-mentioned figure and the others, n=5 means 5 animals used and one sample from one animal.

Effect on paired atrial preparations……

Reply: Tracings indicating the concentration dependent cardio depressant response of TdCr has been added. The observed average initial value of the spontaneous frequency of contraction of the isolated preparation of paired atria is 110 to 130 beats per minute.

Figure 9: FOC is the abbreviation of?

Reply: FOC is the abbreviation used for Force of contraction, added in the list of abbreviations.

Discussion:

Could the authors compare ……… tissues identical?

Reply: KATP are heterogenous in nature. Authors observed almost similar response in the used smooth muscles i.e. jejunum, trachea and the aorta but inhibitory effect on cardiac muscle was not prominent. Previous studies have shown that the cardiac inhibitory effects of cromakalim occurs at concentration, 30-100 folds greater than that causing smooth muscle relaxation (Grossett and Hick,1986; Osterrieder,1988). Role of KATP channels under physiological condition of heart is not well marked (Tong et al., 2006) moreover there is an existence of different isoforms of KATP channels in cardiac and smooth muscles which may explain the relatively less inhibitory effect of TdCr on cardiac muscles (Moreau et al., 2000).

Line 405: the description………………. Its regulation?

Reply: Role of MLC phosphatase in muscle relaxation has been added in manuscript as suggested by respected reviewer.

On the graphical……

Reply: Thank you for the suggestion. Correction has been made.

The authors have characterized………. KATP opener?

Reply: As the main aim was to evaluate the pharmacological activities with possible mechanism, so only crude extracts, and its dichloromethane and aqueous fractions were investigated. Further, characterization can be done, and compounds can be investigated in future perspective.

What are the immediate perspectives…….?

Reply:

In future, pure compound can be isolated and investigated for various activities which can be used as a lead component for drug discovery. This study was performed on the basis of available resources in our laboratory. Electrophysiological studies can be performed on pure compound in future.

Reviewer 3 Report

The current paper by Imtiaz et al. titled as “Potential involvement ATP-dependent potassium 2 channel opening mechanism in smooth muscle 3 relaxant prospective of Tamarix dioica Roxb.” discusses the pharmaceutical role of Tamarix dioica, a domestic herb, on the ATP-dependent potassium channels of smooth muscles, in vivo and ex vivo. This group has found that the Tamarix dioica extracts can affect the K+ channels and have a relaxing effect on the smooth muscles via closure of the Ca2+ channels. The manuscript has been prepared well, however, there are some general and specific comments required to be addressed.

General comments

Grammatical revision is required. There are several sentences with either grammatical flaws or punctuation marks. The introduction did not address the previous work(s) on the topic, nor the gap of knowledge in the field. Clear hypothesis and reasoning is required. The catalog number of the chemicals in Materials and Methods section is required. Some of the abbreviations should be revised, specifically in Materials and Methods section. The word “thrice” is outdated. It is advised to be replaced by “in three replications”. In Materials and Methods section, molarity of the reagents and extracts should be replaced by the volumes The word “EX-vitro” in line 157 should be replaced by “Ex-vivo”. The words rats and rabbits are continuously used without considering the consistency in the manuscript. It causes confusion. Consistency in the manuscript should be addressed; “Kreb’s solution” in several positions used as “kreb solution”. Instead of “killing”, it is advised to use “euthanizing”. The resolution of the figures is not the same. It is advised to use figures with higher resolution.

Specific comments

How was the number of animals per group defined? Was there any power analysis to support the statistical requirements for the in vivo study? Representation of the results the way showed in figure 2 is confusing. the horizontal axis should change to the different groups, and the legend should be the dosage. The units are not the same, so there is no point to show this way. There is a gap between the in vivo study and ex vivo study. How can the results from the in vivo study support results from the ex vivo study when two different animals were used? It has not been clearly discussed why the effect of TdCr in higher concentration of K+ has been eliminated.

Author Response

Dear Sir                               

I am highly thankful to the reviewers for their precious time and invaluable comments for improving manuscript ID: biomolecules-631569 entitled, “Potential involvement of ATP-dependent potassium channel opening mechanism in smooth muscle relaxant prospective of Tamarix dioica Roxb. ”. We have carefully addressed all the comments which really helped in improving the manuscript. We have removed typo and grammar errors.The corresponding changes and refinements made with red color in the revised paper are summarized in our response below.

REVIEWER 3:

Major comments:

Replies:

The extraction and fractionation has been done following the previously established methods. References has been added. Statistical analysis has been added at the end of the section of Methods Future perspective has been added in conclusion Statistical analysis for ex vivo studies has been added

Minor comments:

The suggested change in the title has been made.

Reviewer 4 Report

General comments:

Imtiaz et al have performed in vivo and ex vivo studies in order to evaluate the potential health friendly attributes (antidiarrheal, spasmolytic, bronchodilator, vasorelaxant…) of the shrub Tamarix dioica.

Major comments:

1) In the Methods section (page 2), there is a detailed description of the method for extraction and fractionation. Was this method created de novo or based on previously published protocol? If based on the literature, please indicate the source.

2) There is a missing information at the end of the Methods section about the statistical analysis and how are the numerical data presented. Please include this, despite the fact that such information exists in the figure legends.

3) In general, there are a plethora of different and potentially bioactive compounds detected in the TdCr. And, their potential and already known effects were nicely discussed in the manuscript. However, there has to be mentioned that the future studies should profoundly analyze the potential effects of all extracted molecules in different combinations to be sure what is it in this small tree that makes it so useful for several health disorders. Please include this concept in the conclusion or at the end of discussion.

4) For all ex vivo experiments, there were no statistical tests performed to compare different curves. Please use the proper statistical test(s) and indicate the significant changes for all ex vivo derived data.

Minor comments:

1) Please correct the title of the manuscript. The missing word is “of”: “Potential involvement of ATP-dependent…”

Author Response

Dear Sir                               

I am highly thankful to the reviewers for their precious time and invaluable comments for improving manuscript ID: biomolecules-631569 entitled, “Potential involvement of ATP-dependent potassium channel opening mechanism in smooth muscle relaxant prospective of Tamarix dioica Roxb. ”. We have carefully addressed all the comments which really helped in improving the manuscript. We have removed typo and grammar errors.The corresponding changes and refinements made with red color in the revised paper are summarized in our response below.

REVIEWER 4:

General comments

Grammatical revision is required. There are several sentences with either grammatical flaws or punctuation marks.

Reply: Authors checked the manuscript for grammatical and punctuation marks as per suggested.

The introduction did not address the previous work on the topic, nor the gap of knowledge on the field. Clear hypothesis and reasoning is required.

Reply: The plant is famous for its traditional uses already mentioned in the manuscript

{T. dioica is useful in treatment of diarrhea,  and dysentery, in  inflammation, for cold, fever, flu, cough and as astringent, in burns, leucodermic [5] and is styptic in nature [6]. Paste of dried bark and leaves is useful on wounds [7,8]. T. dioica is active against various serious bacterial and viral infections like tuberculosis, leprosy, gonorrhea, ringworm, polio and measles [9, 10]} and also include the previous work done on it;

{Previous studies suggested the presence of remarkable antifungal activity [11] potent ulcer protective and anti-inflammatory activities in T.dioica [12]. T.dioica (leaves) showed significant activities as hepatoprotective, anti-oxidant/free radical scavenger [13, 14], cytoprotective against gastric epithelial cell damage [5] and antimicrobial [15].  Furthermore, some species of genus Tamarix have been previously reported to have cardio-protective potential e.g. Tamarix aphylla [16]}.Hypothesis has been added in introduction as per suggested by respected reviewer.

The catalogue number of the chemicals in material and method section has been required.

Reply: The catalogue numbers of the important chemicals used in our laboratory has been enlisted here.

Chemical Name

Catalogue No.

CAS no.

Company Name

Methanol

34860

67-56-1

Sigma Aldrich

Dicholoromethane

L090000 

75-09-2

Sigma Aldrich

Ach

A6625

60-31-1

Sigma Aldrich

CCh

212385-M

51-83-2

Sigma Aldrich

Doxazosin

D9815

·         77883-43-3

Sigma Aldrich

Glibenclamide

G0639

10238-21-8

Sigma Aldrich

Loperamide HCL

L4762

34552-83-5

Sigma Aldrich

Phenylephrine

BP284

61-76-7

Sigma Aldrich

Potassium Chloride

409316

7447-40-7

Sigma Aldrich

Cromakalim

C1055

94470-67-4

Sigma Aldrich

Some of the abbreviations should be revised, specifically in material and method section.

Reply: Authors have tried to revise and correct all the used abbreviation

The word “Thrice” is outdated…..

Reply: The word thrice has been replaced by “in three replications” in the manuscript.

In material and method section, molarity should be replaced by the volumes.

Reply: By definition, molarity is also used to indicate volumes. And most commonly used unit in research and research articles.

“In chemistry, molarity is a concentration unit, defined to be the number of moles of solute divided by the number of liters of solution.”

The word “Ex-vitro” in line 157 has been replaced by “Ex-vivoThe word rats and rabbits are continuously used without considering the consistency….

Reply: The use of words “rats and rabbits” has been reduced by rephrasing the sentences.

The word “Krebs solution” has been checked and corrected. The word “killing” has been replaced with “euthanizing”. The figures has been adjusted as per suggestion.

SPECIFIC:

How was the number of animals defined? Was there any power analysis…..

Reply: 5 animals were used in every group of in-vivo study. Statistical analysis has been added at the end of materials and methods.

Representation of the results showed in figure 2…..

Reply: In figure 2, bars in graph indicating different treated groups in the study, X-axis indicating the dosage or treatment received. Only control group received normal saline (unit used is mL/kg) while the all other group received either the standard or the test material and the unit used is mg/kg. Authors edited the figure to make it clearer. 

There is a gap between the in-vivo study and ex-vivo study……

Reply: Initially, we performed in vivo experiments to check the anti-diarrheal potential of Tamarix dioica, then used isolated tissues to confirm and explore the possible mechanism. Previous studies suggested that rabbit jejunum is considered as an appropriate tissue preparation to assess the contractile and/or relaxant effects because of spontaneous rhythmic contractions (Mehmood et al., 2011). While, Rodents both rats and mice are the most common used experimental animals (Ghayur and Gilani, 2005; Meiti et al., 2009; Nwidu et al., 2011).

It has been clearly discussed why the effect of TdCr in higher concentration of K+ has been eliminated.

Reply: Present study indicated that TdCr relaxed the low K+(25 mmol/L) induced contraction with great potency as compared to the high K+(80 mmol/L) induced contractions. Substances that selectively inhibited the contractions provoked by K+(25 mmol/L) are denoted as potassium channel activator/opener Whereas, on other side, substances inhibited both the contractions provoked by K+(25 & 80 mmol/L) are termed as Ca+2 channel blockers. These experiments effectually differentiate between the potassium channel openers and calcium channel blockers from mechanistic viewpoint. This point is already discussed in the discussion.

REFERENCES:

Ashour,O.M; Nagy, A.A; Abdel-Naim, A.B; Abdallah, H.M; mohamadin, A.M; Abdel-Sattar, E.A.  Evaluation of the Potential cardioprotective Activity of Some Saudi Plants against Doxorubicin Toxicity. Z. Naturforsch. 2012, 67 c, 297 – 307.

Ghayur, M.N., Gilani A.H., 2005. Pharmacological basis for the medicinal use of Ginger in gastrointestinal disorders. Digestive Diseases and Sciences 50(10), 1889–1897.

Grossett, A., Hicks, P.E., 1986. Evidence for blood vessel selectivity of BRL34915. British Journal of Pharmacology 89, 500P.

Mehmood, M.H., Siddiqi, H., Gilani, A., 2011. The antidiarrheal and spasmolytic activities of Phyllanthus emblica are mediated through dual blockade of muscarinic receptors and Ca2+ channels. Journal of Ethnopharmacology 133(2), 856-865.

Meite, S., N’Guessan, J.D., Bahi, C., Yapi, H.F., Djaman, A.J., Guina, F.G., 2009.Antidiarrheal activity of the Ethyl Acetate Extract of Morinda morindoides in Rats. Tropical Journal of Pharmaceutical Research 8(3), 201-207

Moreau, C., Jacques, H., Prosa, A.L., Dhahan, M., Vivaudou, M., 2000. The molecular basis of the specificity of action of KATP channel opener. The EMBO Journal 19, 6644-6651.

Nwidu, L.L., Essien, G.E., Nwafor, P.A., Vilegas, W., 2011. Antidiarrheal mechanism of Carpolobia lutea leaf fractions in rats. Pharmaceutical Biology 49(12), 1249-1256

Osterrieder, W., 1988. Modification of K+ conductance of heart cell membrane by BRL 34915. Naunyn-Schmiedeberg’s Archives of Pharmacology 337, 93-97.

Tong, X.Y., Porter, L.M., Liu, G.X., Chowdhury, P.D., Srivastava, S., Pountney, D.J., Yoshida, H., Artman, M., Fishman, G.I., Cindy, Y.U., Iyer, R., Morley, G.E., Gutstein, D.E., Coetzee, W.A.,2006. Consequences of cardiac myocyte-specific ablation of KATP channels in transgenic mice expressing dominant negative Kir6 subunits. American Journal of Physiology - Heart and Circulatory Physiology 291(2), H543-H551

Thank you once again for your valuable comments. I am available if there are any further queries.

--

Highest Regards

Dr. Andrea Elena Neculau

Round 2

Reviewer 3 Report

Comments are sufficiently addressed.